# Bonding of Carbon Steel Bars in Concrete Produced with Recycled Aggregates: A Systematic Review of the Literature

**Elvys Dias Reis** *[ID], **Henrique Comba Gomes** [ID], **Rogério Cabral de Azevedo** [ID], **Flávia Spitale Jacques Poggiali** [ID] and **Augusto Cesar da Silva Bezerra** [ID]

Federal Center for Technological Education of Minas Gerais, Belo Horizonte 30421-169, Minas Gerais, Brazil
* Correspondence: elvysreis@yahoo.com.br

**Abstract:** Civil construction is essential for the world economy and the largest generator of construction and demolition waste (CDW), mainly due to a lack of planning, technological control, and restoration execution, among other factors. While efforts are made to minimize this waste generation, one possible application for CDW is its incorporation into Portland cement-based materials as recycled aggregates, in partial or total replacement of natural aggregates. However, for CDW use to be feasible, the structure performance and safety must be assured, and the adherence between concrete and reinforcement bars, in this context, is a fundamental mechanism. With this perspective, this paper aims to investigate the influence of recycled aggregate on steel–concrete bonding. To this end, the SREE (Systematic Review for Engineering and Experiments) method was employed as a novelty, including a methodology quality analysis, to search and analyze relevant scientific articles published in the last ten years. The results revealed that the use of CDW as recycled aggregates in concrete worsens the steel–concrete bonding, and that ribbed steel bar seems to be the best option when employed in RC structures built with CDW-concrete, although the bar diameter and the anchorage length still need further investigations, and that CDW-concrete's use can significantly contribute to reducing the emission of greenhouse gases and to capturing $CO_2$ from the atmosphere. Therefore, further investigations should focus on the real influence of recycled aggregate type and replacement content, bar diameter, anchorage length, and CDW's potential to capture $CO_2$.

**Keywords:** anchorage length; bibliographic analysis; bond strength; cementitious materials; pull-out test; recycled aggregates; SREE; steel bars

## 1. Introduction

The construction industry, despite being prominent in the economy, generates negative impacts, both by the extraction and consumption of natural resources and by the modification caused to the environment due to the waste generated in construction and demolition. In this aspect, construction and demolition waste (CDW) represent the largest mass quantity of urban solid waste [1]. In large and medium-sized Brazilian cities, CDW generation corresponds to approximately 41 to 71% of the total solid waste generated, by mass [2]. Moreover, they are usually dumped in inappropriate places, and in 2012 only 6% of what was generated was recycled [3]. Data released by the Brazilian Association of Public Cleaning and Special Waste Companies in 2022, however, indicate that the recycling rate has decreased, and is currently only 4% [4]. This 4% recycling rate in Brazil is well below the rate recorded in countries with similar income ranges and economic development, such as Chile, Argentina, South Africa, and Turkey, which recycle about 16% of the waste they produce, according to the International Solid Waste Association (ISWA), and far below rates such as Germany, which recycles approximately 67% of its waste [5].

Moreover, although concrete emits less carbon dioxide ($CO_2$) than steel, wood, glass, and other building materials [6], most conventional concretes used by the construction industry contain Portland cement, whose production process emits large amounts of $CO_2$

into the atmosphere [7]. With this perspective, CDW use as recycled aggregates in concrete production has been an eco-friendly alternative. Some studies have even pointed out that concrete produced with CDW as recycled aggregates (CDW-concrete) can capture $CO_2$ from the atmosphere [8,9]. In this sense, Figure 1 shows the growing number of studies involving the production of CDW-concrete.

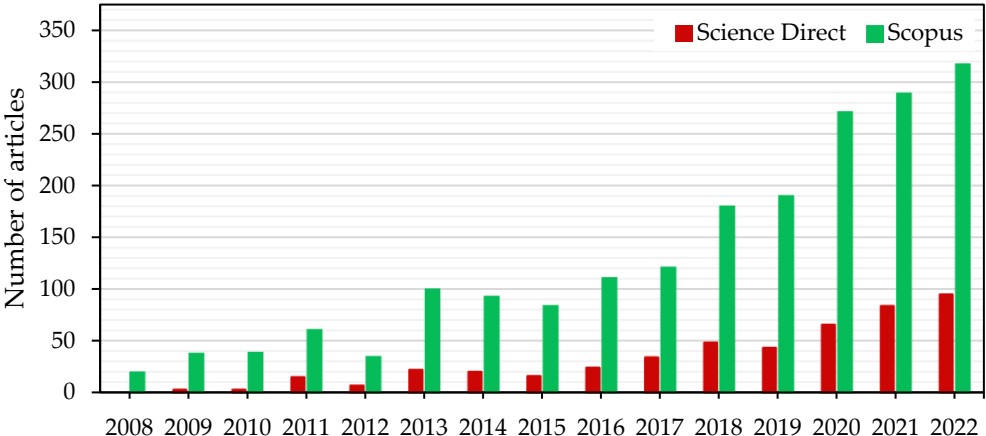

**Figure 1.** Evolution of research on concrete produced with CDW as recycled aggregates.

When concrete performs a structural function, i.e., in reinforced concrete (RC) structures, it must guarantee some specifications concerning its mechanical strength, such as excellence or virtually perfect bonding between the cementitious material and the reinforcing bars. The bond between steel and concrete has been studied since the first applications of RC in civil construction, being linked to the evolution of this construction system itself. Although there is more knowledge about this mechanism nowadays, the development of new construction materials makes the subject increasingly investigated. Some of its aspects are well established, but there are still several knowledge gaps to be filled in future research [10].

The steel–concrete adherence mechanism is one of the most important for RC structures because the two materials must act together to bear the internal forces. The adherence anchors the reinforcement in the concrete and prevents its slipping in the segments between cracks, limiting their opening [11].

In this context, CDW-concrete has been widely studied regarding its bonding with carbon steel bars [12,13] due to the large amount of CDW generated by the construction industries. However, CDW use as recycled aggregates is still low in concrete production because they are quite heterogeneous, and there is a lack of incentives and standardization for their use [14], besides presenting mechanical properties with lower values compared to natural ones [15]. Thus, they end up being sent directly to landfills, which seems to be the simplest alternative [1]. However, these landfills produce gases like methane ($CH_4$) that aggravate the greenhouse effect and contribute to increased global warming [16,17].

According to the Brazilian standard NBR 15116 [18], in concrete production, the replacement of natural aggregates by artificial ones should be limited to 20% of the total mass of aggregates, considering as a source only concretes of aggressiveness classes I and II of the NBR 6118 standard [19]. This limitation may be due to the great variability and quality of CDW. The possibility of applying CDW in concrete production justifies the need for further investigations on the use of recycled aggregates, especially for the manufacture of materials for structural purposes, the study of steel–concrete bond strength being a relevant alternative in this regard.

*Research Significance*

RC structures are globally widespread and when compared to other structural systems, the availability of the constituent materials (concrete and steel) and the ease of application

explain their wide use in various types of construction [20]. On the other hand, concrete production is responsible for about 7% of global $CO_2$ emissions, since it is the most consumed material in the world [21]. In this sense, the sustainability and safety of RC structures deserve increasing attention from the technical and scientific community, due to the high generation of CDW and the occurrence of accidents in these structures, sometimes caused by deficient anchorage length or steel–concrete loss of adherence.

In this context, bond strength is necessary to ensure an adequate safety level, control structural behavior, and ensure sufficient ductility between steel and concrete [22]. This means that a weak or insufficient steel–concrete bond strength can cause the slipping of reinforcement steel and reduce the flexural strength and deformation capacity of the RC structure [23], which could lead to excessive deflection, cracking, or collapse of the entire structure [24]. In addition, perfect steel–concrete bonding is generally an assumption for the design of RC structures; however, this is valid only for small slip values. In practice, reinforcing steel bars are subject to displacement relative to concrete, hence the need to know the relationship between bond stress and displacement to more accurately evaluate the behavior of a structure. Some studies propose models for the bond stress vs. displacement curve, as well as expressions for estimating the maximum bond stress of differences between steel and concrete [25–27].

Given the importance of steel–concrete bond strength for the performance and safety of RC structures and the use of CDW to mitigate environmental impacts, scientific research should provide consistent results for society, contribute to discoveries, and enable the development of innovative products and practical applications. Therefore, this manuscript presents the state-of-the-art on the adherence of steel bars in CDW-concrete, considering the bond test, replacement content, water/cement ratio (w/c), bar type, diameter, and anchorage length, to answer the following questions: (i) Which replacement content led to the best results in the bond strength of carbon steel bars in CDW-concrete? (ii) What are the main characteristics of the carbon steel bars and the anchorage length employed in the study of adherence with CDW-concrete? (iii) What is the environmental perspective regarding using CDW recycled aggregates in RC structures? To this end, a bibliographic analysis and systematic review of the literature were conducted to discuss future directions in the study of carbon steel bar adherence in CDW-concrete. In summary, Figure 2 presents the main steps of this work.

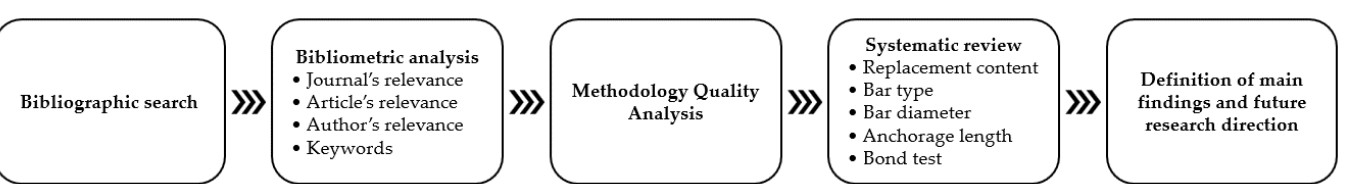

**Figure 2.** Main stages of this research.

## 2. Materials and Methods

### 2.1. Systematic Review

To perform the systematic review, the SREE (Systematic Review for Engineering and Experiments) [28] method was employed as a novelty, which consists of an adaptation and improvement of the ProKnow-C (Knowledge Development Process-Constructivist) method [29] through the inclusion of a methodology quality analysis of the scientific papers, as described later.

In the first step, the search parameters were defined. The keywords "bonding", "concrete", "steel", and "recycled aggregate" were searched for in the title, abstract, and keywords of the papers in the literature. The search was conducted on 9 June 2022, and considered any article type published between 2012 and 2021 and present in the Science Direct, Scopus, or Scielo databases, all indexed by the CAPES [30] website. This search resulted in 664 manuscripts. In the second phase, 621 papers were extracted from the databases and entered into a reference manager. This stage only considered full manuscripts

published in journals. Considering these, 19 were removed for being duplicates, i.e., indexed in at least two of the three databases chosen. Thus, 602 papers remained in the third stage, of which 78 presented titles aligned with the research theme in the fourth stage. In the fifth, 47 articles were selected because they presented good scientific recognition, corresponding to 85% of the citations of the portfolio selected until the previous stage. Then, 15 recent papers (published in 2020 and 2021) which were, therefore, papers with few citations and that had not gone through the previous filters, were added to the portfolio, which now had 62 papers, in the sixth stage. Considering these, 22 manuscripts were chosen for having abstracts aligned with the research topic, in the seventh stage, of which it was verified that two were not fully available, leaving 20 articles in the portfolio, in the eighth stage. In the ninth and last stage, all 20 articles went through a full reading and were fully aligned with the theme in question. The entire process can be seen in Figure 3.

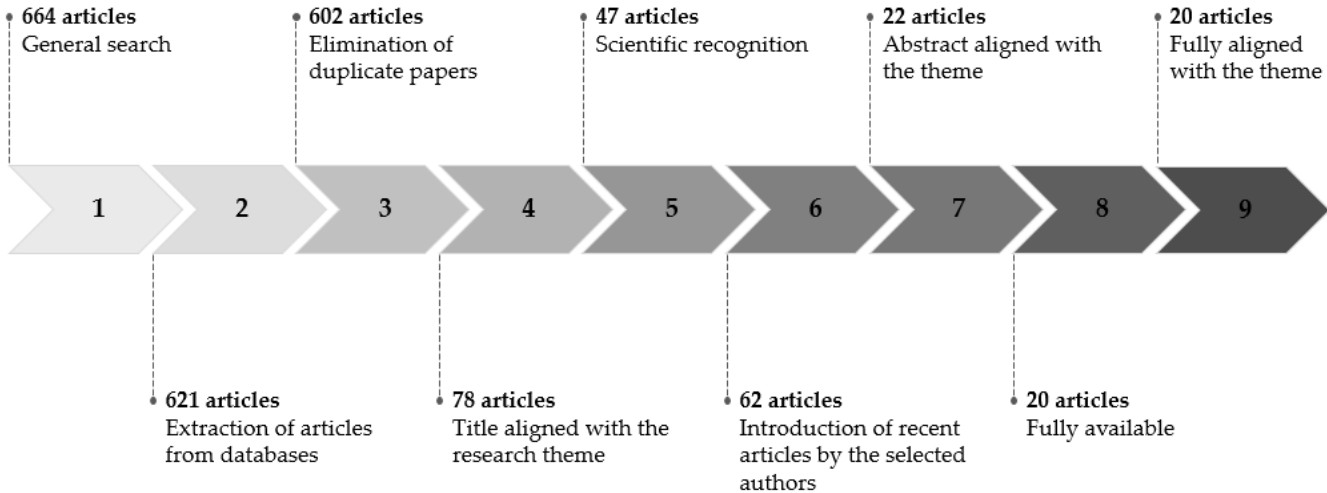

**Figure 3.** Stages to select the bibliography.

Table 1 lists the articles that make up the selected bibliography, detailing their title, journal, and publication year.

**Table 1.** Details of selected manuscripts.

| Reference | Title | Journal | Year |
|:---:|:---:|:---:|:---:|
| [31] | Tests and simulation of the bond-slip between steel and concrete with recycled aggregates from CDW | *Buildings* | 2021 |
| [32] | Bond of epoxy-coated steel bars to seawater sea sand recycled concrete | *Structures* | 2021 |
| [33] | Bond performance of deformed steel rebars in HSC incorporating industrially produced recycled concrete aggregate | *Materials and Structures* | 2021 |
| [34] | Bond strength behavior for deformed steel rebar embedded in recycled aggregate concrete | *Journal of Engineering and Technological Sciences* | 2021 |
| [35] | The study on bond-slip constitutive model of steel-fiber high-strength recycled concrete | *Structures* | 2021 |
| [36] | Effect of quality of recycled aggregates on bond strength between concrete and embedded steel reinforcement | *Journal of Sustainable Cement-Based Materials* | 2020 |
| [37] | Bond behavior between deformed steel bars and recycled aggregate concrete after freeze-thaw cycles | *Construction and Building Materials* | 2020 |
| [38] | Analytical investigation in bond of deformed steel bars in recycled aggregate concrete | *Journal of Sustainable Cement-Based Materials* | 2020 |

**Table 1.** *Cont.*

| Reference | Title | Journal | Year |
|---|---|---|---|
| [39] | Bond behavior of steel bar embedded in recycled coarse aggregate concrete under lateral compression load | *Construction and Building Materials* | 2017 |
| [40] | Investigation of compressive bond behavior of steel rebar embedded in concrete with partial recycled aggregate replacement | *Structures* | 2016 |
| [41] | Steel–concrete bond behaviour of self-compacting concrete with recycled aggregates | *Magazine of Concrete Research* | 2016 |
| [42] | Bond strength prediction for deformed steel rebar embedded in recycled coarse aggregate concrete | *Materials & Design* | 2015 |
| [43] | Bond behavior between steel bar and recycled aggregate concrete after freeze–thaw cycles | *Cold Regions Science and Technology* | 2015 |
| [44] | Bond strength of deformed steel bars in high-strength recycled aggregate concrete | *Materials and Structures* | 2015 |
| [45] | Bond behavior between steel reinforcement and recycled concrete | *Construction and Building Materials* | 2014 |
| [46] | Evaluation of the bond behavior of steel reinforcing bars in recycled fine aggregate concrete | *Cement and Concrete Composites* | 2014 |
| [47] | Bond behaviour between recycled aggregate concrete and deformed steel bars | *Materials and Structures* | 2014 |
| [48] | Bond behaviour of deformed steel bars embedded in recycled aggregate concrete | *Construction and Building Materials* | 2013 |
| [49] | Structural reliability of bonding between steel rebars and recycled aggregate concrete | *Construction and Building Materials* | 2013 |
| [50] | Bond performance of deformed steel bars in concrete produced with coarse recycled concrete aggregate | *Canadian Journal of Civil Engineering* | 2012 |

*2.2. Bibliographic Analysis*

The bibliometric analysis was conducted considering: (i) the journal's relevance within the selected portfolio, i.e., how many articles were published in each of them; (ii) the papers' scientific recognition, through the number of citations in Google Scholar [51], the journal classification in the 2013–2016 quadrennium, according to CAPES [30] (this classification divides the publications into groups: A1, A2, B1, B2, B3, B4, B5, and C, from the best to the worst classification), and the JCR (Journal Citation Reports) impact factor, according to the Web of Science (WoS) database; (iii) the author's relevance, i.e., in how many papers they appear as authors or co-authors within the selected portfolio; and (iv) the most frequent keywords.

2.2.1. Journals' Relevance and Papers' Scientific Recognition

The 20 manuscripts were published in 11 different journals. Figure 4 shows the number of times each paper was cited from the publication date until 17 October 2022, according to Google Scholar [51]. In addition, the classification according to CAPES [30] was verified, as well as the journal's JCR impact factor, as shown in Table 2.

The analysis of Figure 4 shows that the papers published longer tend to have a greater number of citations, as expected, while the verification of Table 2 adds that the journal's quality has more influence on the article's scientific recognition. In other words, better-ranked journals and, consequently, with a greater impact factor, commonly have more outstanding visibility in the scientific community and tend to present robust theoretical, numerical, and/or experimental studies. From the selected portfolio, the journals *Cement and Concrete Composites* (JCR = 9.930), *Materials & Design* (JCR = 9.417), and *Construction and Building Materials* (JCR = 7.693) are the most relevant.

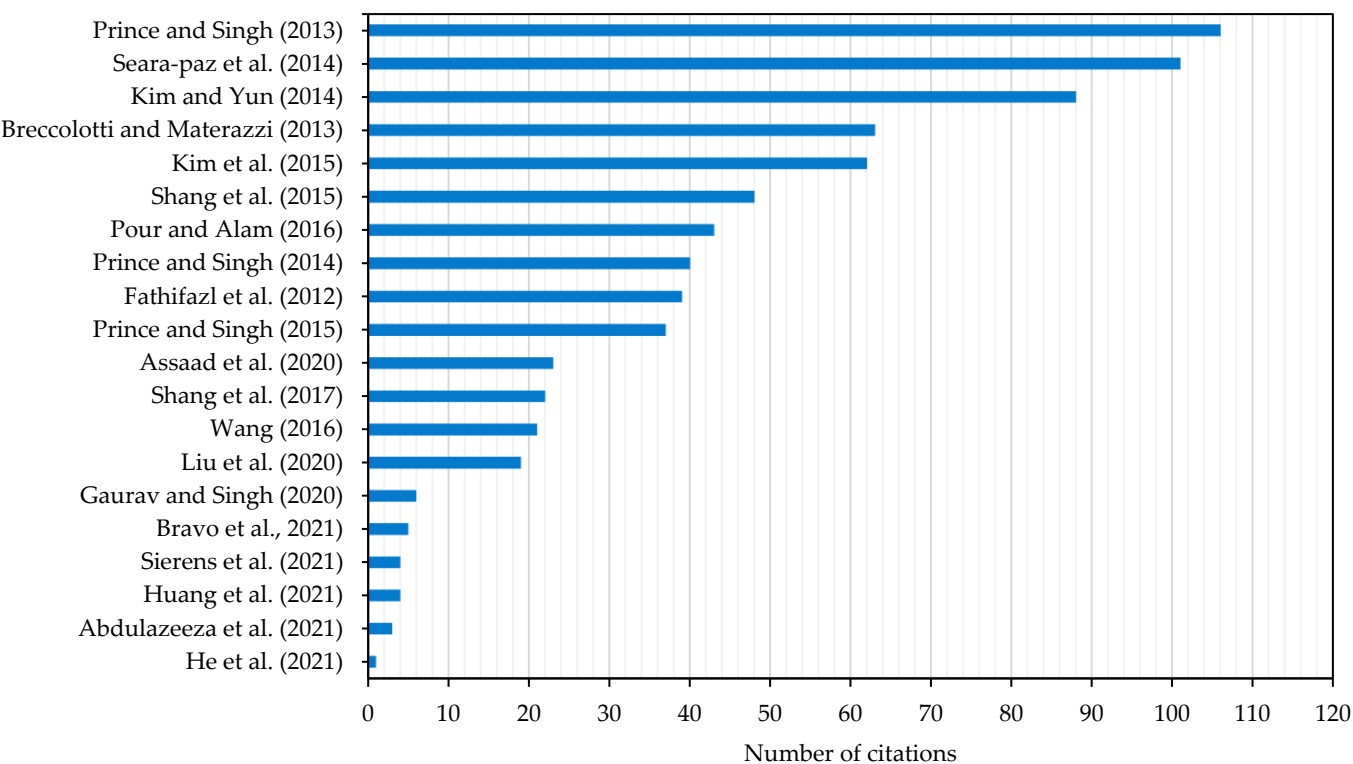

**Figure 4.** Relevance of the selected articles.

**Table 2.** Classification of the journals by article.

| Reference | Journal | Classification | JCR |
|---|---|---|---|
| [46] | *Cement and Concrete Composites* | A1 | 9.930 |
| [42] | *Materials & Design* | A1 | 9.417 |
| [37] | *Construction and Building Materials* | A1 | 7.693 |
| [39] | *Construction and Building Materials* | A1 | 7.693 |
| [45] | *Construction and Building Materials* | A1 | 7.693 |
| [48] | *Construction and Building Materials* | A1 | 7.693 |
| [49] | *Construction and Building Materials* | A1 | 7.693 |
| [36] | *Journal of Sustainable Cement-Based Materials* | - | 5.328 |
| [38] | *Journal of Sustainable Cement-Based Materials* | - | 5.328 |
| [43] | *Cold Regions Science and Technology* | - | 4.427 |
| [33] | *Materials and Structures* | A1 | 4.285 |
| [44] | *Materials and Structures* | A1 | 4.285 |
| [47] | *Materials and Structures* | A1 | 4.285 |
| [32] | *Structures* | B4 | 4.010 |
| [35] | *Structures* | B4 | 4.010 |
| [40] | *Structures* | B4 | 4.010 |
| [31] | *Buildings* | - | 3.324 |
| [41] | *Magazine of Concrete Research* | A2 | 2.460 |
| [50] | *Canadian Journal of Civil Engineering* | A2 | 1.771 |
| [34] | *Journal of Engineering and Technological Sciences* | - | - |

### 2.2.2. Author's Relevance

The author's relevance considers the number of papers in which they appear as an author or co-author. The most frequent authors were the Indian Bhupinder Singh and M. John Robert Prince, with four and three papers, respectively.

CDW recycled aggregates have a wide composition diversity, varying according to the place of origin [14]. In the selected bibliography, Asian countries are predominant, such as China, India, South Korea, Lebanon, and Iraq, regions with a large production of demolition waste because of the frequent involvement with wars in that region. In the Lebanon–Israel war in 2006, for instance, six million m$^3$ of demolition waste was generated, most of it ending up in inadequate landfill sites throughout the country [52]. It is also worth mentioning the Europeans, such as Portugal, Belgium, Spain, and Italy, and North America, Canada.

### 2.2.3. Keywords

In this analysis, 65 different keywords were identified in the 20 manuscripts, among which were: "recycled aggregate concrete", with thirteen occurrences; "bond strength", with eight; "pull-out test", with five; "bond", with four; and "coarse recycled concrete aggregate", with three. Figure 5 shows the clustering of keywords in the portfolio, in which one can identify other terms commonly used in studies about the adherence of steel bars in concretes produced with recycled aggregates.

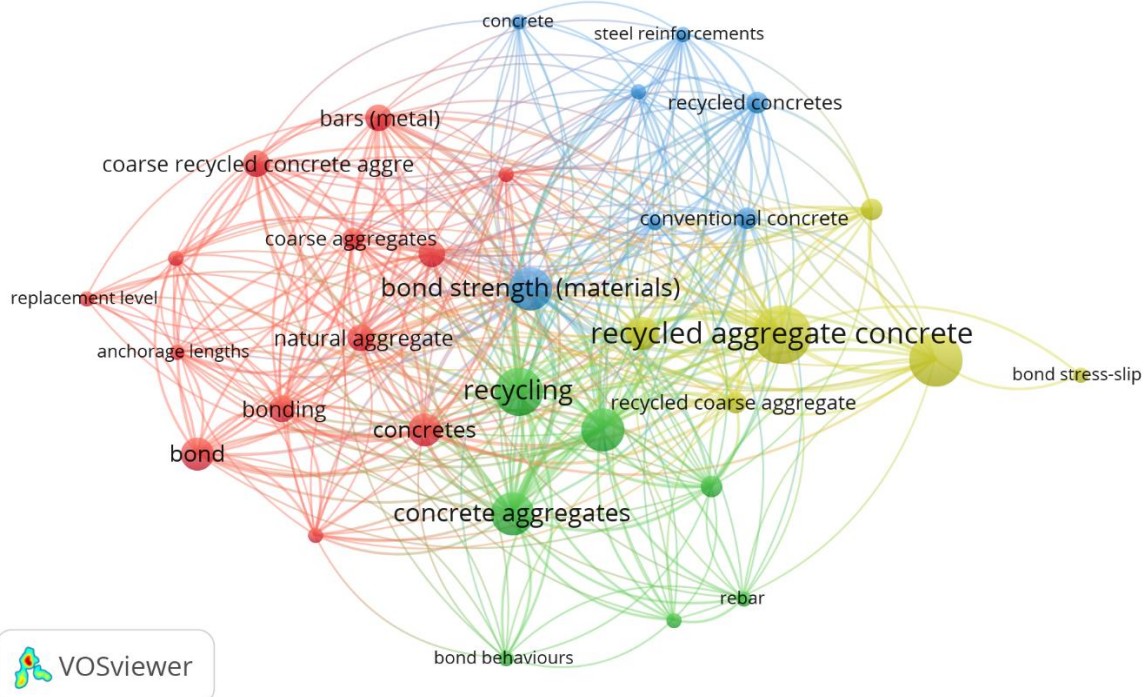

**Figure 5.** Clustering of keywords from the selected bibliography.

### 2.3. Methodology Quality Analysis

The SREE (Systematic Review for Engineering and Experiments) [20] method was used to evaluate the methodological quality of the selected studies. This method considers the following criteria:

(1) Randomization: the random error, also known as variability or random variation, reflects the influence of uncontrolled factors on experiments and occurs mainly due to small fluctuations in the instruments, the environment, or the way measurement is read, and also by small differences in the process of producing the samples (materials, dosages, temperature, and human influences, among others). Randomization aims to distribute the random error over all samples, preventing it from influencing one more than another.

(2) Analysis: the use of descriptive statistics (means, standard deviation, and coefficient of variation) to compare samples characterizes only the results of the experiment, that is, the samples generated by the experiment. The use of inferential statistics (hypothesis tests) allows for describing and making generalizations about the population, that is, allows for characterizing the method that generated the samples.

(3) Comparison: the comparison allows the best results obtained in the experiment to be highlighted, and all scientific research uses it. However, comparison can be restricted to the scope of the experiment itself (analysis) or include other research with similar objectives, allowing the reader a more comprehensive view of the state-of-the-art related to the theme. The choice of research for comparison can be made by the researcher or obtained through a systematic review, and the comparison itself can be conducted through descriptive or inferential statistical tools.

In summary, Table 3 presents the items considered in this evaluation.

**Table 3.** Items for the methodology quality analysis of a paper according to the SREE method.

|   | Item | | Description |
|---|------|---|-------------|
| I | Randomization | | Evidence of randomness in the production and testing of the samples involved |
| II | Analysis | Basic | Use of mean and standard deviation to characterize the sample elements |
|   |   | Statistic | Use of inferential statistics to characterize the method |
| III | Comparison | Basic | Comparison with reference elements (or samples) |
|   |   | Median | Comparison with similar studies without indication of origin (systematic review) |
|   |   | Advanced | Comparison with similar studies from a systematic review |
|   |   | Statistic | Use of inferential statistics for comparison with systematic review studies |

Figure 6 shows the number of articles in which each item proposed for methodology quality analysis was checked.

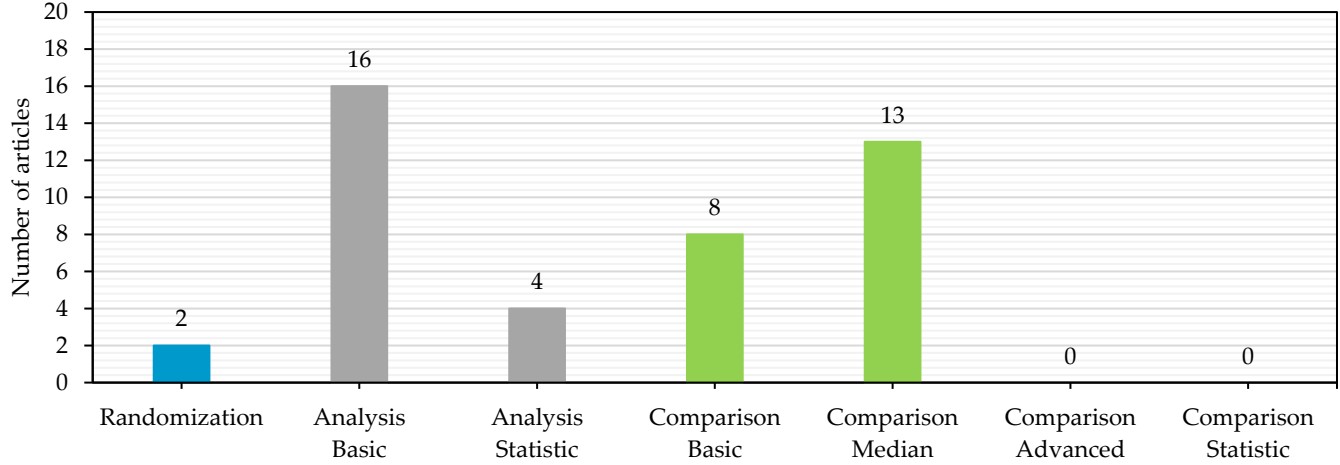

**Figure 6.** Items checked in the methodology quality analysis.

As shown in Figure 6, only two articles (10%) evidenced randomization in the sample preparation or tests performed in the study about the bond strength of steel bars in CDW-concrete. Evaluating the sample's randomization is essential to confer statistical reliability to the work since it distributes the statistical errors present in laboratory tests, minimizing their impact on the results. Thus, it is unknown if any human or material effect influenced the specimens' production or testing in more than 90% of the studies. Figure 6 also shows that 16 manuscripts (80%) used only the mean and standard deviation, measures of descriptive statistics, to compare the outcomes of the samples for the

treatments used, so the results are restricted to those samples (Analysis-Basic). In the remaining four articles (20%), inferential statistics were used to characterize the method, so the results of the samples can be transferred to the population (Analysis-Statistic). As for the comparison and discussion of the results, finally, it was noted that in eight articles (40%) only the comparison of the samples regarding the treatments with the reference sample was performed (Comparison-Basic), while in the other 13 manuscripts (65%) the results were compared with data from similar studies, but which were not selected from a systematic review (Comparison-Median). Neither "Comparison-Advanced", with studies coming from a systematic review, nor "Comparison-Statistic", with normalized statistical comparisons, was verified in the portfolio.

These data show that, although the topic of steel bar adherence in CDW-concrete has great relevance in the context of sustainability in civil construction, scientific papers must better develop their methodological quality, presenting more complete statistical studies so that they can be extrapolated and compared with other similar works. This comparison is important so that the scientific results may converge to discoveries and enable the development of new products and practical applications.

## 3. Results and Discussion

Table 4 presents a summary of the bibliography, indicating the replacement content and the recycled aggregate granulometry (fine or coarse), bar type, bar diameter, anchorage length, the adherence test performed, and the main conclusions of each article. It allows for answering the research questions presented below.

**Table 4.** Main details and conclusions from the selected bibliography.

| Reference | Replacement Content | Bar Type | Bar Diameter (mm) | Anchorage Length (mm) | Bond Test | Main Conclusions |
|---|---|---|---|---|---|---|
| [31] | 10% FA or CA 50% FA or CA 100% FA or CA | Ribbed | 12.00 | 8 d | Pull-out | • For every 10% of NA replaced with RA, the bond strength decreases by approximately 5%.<br>• The RILEM RC6 standard estimates well the adherence for RAC. |
| [32] | 100% FA | (i) Ribbed (ii) Epoxi-coated and ribbed | (i) 16.00, (ii) 16.00 + 0.17 (epoxi) | 3 d 5 d 8 d | Pull-out | • The influences of sea sand and seawater on the failure mode of the specimen in the pull-out test are negligible.<br>• The bond strength of SSSRC-ECSB is on average 16.5% higher than that of RC-ECSB and 15.3% lower than that of OC-ECSB due to concrete properties. |
| [33] | 10% CA 20% CA 50% CA 100% CA | Ribbed | 16.00 | 3 d | Pull-out | • The incorporation of RA did not affect the steel–concrete bond performance.<br>• The equation proposed by the fib Model Code 2010 fitted well to the curves obtained from the tests, regardless of the amount of RA in the concrete, since the maximum bond stress is used. |

**Table 4.** *Cont.*

| Reference | Replacement Content | Bar Type | Bar Diameter (mm) | Anchorage Length (mm) | Bond Test | Main Conclusions |
|---|---|---|---|---|---|---|
| [34] | 50% CA 100% CA | Ribbed | 12.00, 16.00, 22.00, 25.00 | 5 d 12 d | Push-out | • The bond strength was reduced by up to 13% compared to NAC. <br> • The bonding behavior of RAC was similar to that of NAC. |
| [35] | 30% CA 50% CA 100% CA | Ribbed | 14.00 | 5.7 d | Pull-out | • When the replacement content of NA with RA was kept unchanged, the slippage of the bars increased with the increase of the steel fiber addition content. <br> • When the steel fiber content was kept constant, the bond strength decreased with the increase of the RA replacement content. |
| [36] | 50% CA | Ribbed | 12.00 | 5 d | Pull-out | • The maximum bond strength depended on the quality of the concrete from which the RA originated. <br> • Low-strength RA decreased the bond strength, while high-strength RA increased it. |
| [37] | 30% CA 50% CA | Ribbed | 16.00 | 5 d | Pull-out | • The more freeze-thaw cycles, the lower the steel–concrete bond strength in all samples;· <br> • The bond strength was lower in RAC than in NAC when the number of cycles was fixed. |
| [38] | 50% CA 100% CA | Ribbed | 12.00, 20.00 | 60—500 d | Four-pointflexural | • Similar bond behavior was observed in NAC and RAC, indicating that the force transfer mechanism is the same in both, regardless of the strength level and concrete types. |
| [39] | 30% CA | (i) Smooth (ii) Ribbed | (i) 12.00 (ii) 14.00, 18.00, 22.00 | 5 d | Pull-out | • The tensile strength and bond strength for ribbed steel bars are higher than for plain steel bars under the same lateral compressive force. <br> • The bond stress increases with increasing lateral compressive force. |
| [40] | 30% CA 40% CA 50% CA | Ribbed | 11.00, 16.00, 19.50 | 5 d 10 d | Push-out | • The bond stress was similar for RAC and NAC; <br> • Slip and bond strength increased with an increasing bar cover; <br> • The replacement of 30% of RA showed the best performance both in terms of compressive strength and adherence. |

**Table 4.** *Cont.*

| Reference | Replacement Content | Bar Type | Bar Diameter (mm) | Anchorage Length (mm) | Bond Test | Main Conclusions |
|---|---|---|---|---|---|---|
| [41] | 25% CA 50% CA 100% CA | (i) Ribbed (ii) Smooth | (i) 16.00, (ii) 10.00 | 5 d | Pull-out | • In SCC with RAC, $f_{ck}$ and $f_{ct}$ were greatly influenced by the type of RAC, the source of RAC, and the substitution content. • With pre-treated RAC, the content had little effect. • The adherence decreased greatly when pretreated RAC was replaced by poor-quality RAC obtained from demolished structures. |
| [42] | 30% CA 60% CA 100% CA | Ribbed | 16.00 | 4 d | Pull-out | • Due to the good bond between the aggregate and the mortar matrix, the RAC produced higher pull-out strength than normal concrete. • The bond strength of RAC was affected by the compressive strength, average density, and water absorption of the RA. |
| [43] | 30% CA | (i) Smooth (ii) Ribbed | (i) 12.00 (ii) 14.00, 18.00, 22.00 | 5 d | Pull-out | • Bond strength decreases as freeze-thaw cycles increase. • The drop in bond strength between the smooth steel bar and RAC is greater than that with the ribbed bar as freeze-thaw cycles increase. |
| [44] | 25% CA 50% CA 75% CA 100% CA | Ribbed | 8.00, 10.00 | 5 d | Pull-out | • The bonding mechanism was similar in high-strength RAC and NAC. • The adhesion in RAC increased with increasing substitution content, which was explained by the fracture toughness of RAC. |
| [45] | 25% CA 50% CA 100% CA | Ribbed | 10.00 | 5 d | Pull-out | • The bond strength at 28 days decreases with the increase of the amount of RA, as in conventional concrete. • Concrete with 100% of RAC had bond strength reduced by up to 22% ($w/c = 0.65$) and up to 27% ($w/c = 0.50$). • Age did not interfere much with the results: concrete at 365 had almost no variation in adherence. |
| [46] | 30% FA 60% FA 100% FA | Ribbed | 16.00 | 4 d | Pull-out | • The bond strength was not affected RA use up to 60% replacement. • The adherence to the horizontal pull-out tests was affected by the workability of the concrete rather than the RA content. |

**Table 4.** *Cont.*

| Reference | Replacement Content | Bar Type | Bar Diameter (mm) | Anchorage Length (mm) | Bond Test | Main Conclusions |
|---|---|---|---|---|---|---|
| [47] | 25% CA 50% CA 75% CA 100% CA | Ribbed | 8.00, 10.00 | 5 d | Pull-out | • The bond strength mechanisms are similar in RAC and NAC. • For all RA contents, the trends for 10 mm diameter rebars showed higher adherence than in NAC. • For the same $f_{ck}$, the anchorage lengths of 10 mm rebar in RAC can be conservatively considered the same as for NAC concrete. |
| [48] | 100% CA | Ribbed | 12.00, 16.00, 20.00, 25.00 | 5 d | Beam-end | • A new mixing method has been proposed: the Equivalent Mortar Volume (EVM) dosing method. • The bond strength in RAC provided by the new method is comparable to that in conventional concrete, but 18% to 33% higher than its strength with RAC using conventional methods. |
| [49] | 50% CA 100% CA | Ribbed | 14.00 | 5 d | Pull-out | • The bond strength decreased in RAC, as did the compressive strength, especially when replacing 100% RA. • The fib Model Code 2010 estimates well the stress for RAC, even though it is made for NAC. |
| [50] | 100% CA | Ribbed | 16.00, 30.00 | 250—320 d | Pull-out | • The adherence mechanism was similar in all concretes. • The bond stresses increased with the levels of replacement of RA and the highest values were obtained for 100% replacement of coarse natural aggregate by RAC. • The bond strengths of 12 mm, 16 mm, 20 mm, and 25 mm diameter bars in RCA concrete were higher than those of NAC concrete and this is attributed to the internal curing action of RAC particles. • The bond strengths increased with increasing replacement contents. |

Note: CA—coarse aggregate; ECSB—epoxy-coated steel bars; FA—fine aggregate; $f_{ck}$—compressive strength; $f_{ct}$—tensile strength; NAC—natural aggregate concrete; OC—ordinary concrete; RA—recycled aggregate; RAC—recycled aggregate concrete; RC—recycled concrete; SCC—self-compacting concrete; SSSRC—seawater sea sand recycled concrete.

*3.1. Which Replacement Content Led to the Best Results in the Bond Strength of Carbon Steel Bars in CDW-Concrete?*

It can be observed that several replacement contents have been tested by the authors, both for coarse and fine aggregates. Although most of the papers explicitly reported that

the bond mechanism is similar in CDW-concrete and concrete with natural aggregates, no clear pattern was identified. In addition, compared to conventional concrete, the majority of the articles pointed to a worsening of the bond strength. This mainly stems from reduced compressive strength due to the increased porosity of the CDW-concrete [53,54]. In addition, 17 manuscripts (85% of the bibliography) used the pull-out test, which has good acceptance in the scientific community and presents accurate results, which is in accordance with the literature [10].

Considering the urgent need to properly recycle CDW, it is worth mentioning that several studies have been conducted on the bond behavior of steel bars in CDW-concrete. Bravo et al. [31] numerically and experimentally analyzed the SC bonding with CDW recycled aggregates from several recycling plants in Portugal. The authors focused on the replacement contents and concluded that for every 10% of natural aggregate replaced by the recycled one, the bond stress decreases by approximately 5%. Abdulazeeza et al. [34] investigated the bonding in CDW-concrete through numerical and experimental analyses and the bond strength was reduced by 13% when using recycling aggregates. Romanazzi et al. [55], in turn, studied steel–concrete adherence in concrete with partial aggregate replacement by waste tire rubber (RuC) to evaluate the impact of the content of volumetric substitution of aggregates by rubber waste. The authors noticed a drop in bond strength of up to 20% when replacement levels of natural aggregates greater than 12% are employed, which is justified by a drop of up to 37% in compressive strength compared to conventional concrete. Although being a sustainable alternative, studies have shown that rubber particles reduce the bond strength as the friction between steel and concrete decreases with rubber aggregates' use [56].

### 3.2. What Are the Main Characteristics of the Carbon Steel Bars and the Anchorage Length Employed in the Study of Adherence with CDW-Concrete?

As can be seen in Table 4, carbon steel bars with a ribbed surface were employed in all of the studies, while those with a smooth surface were used in three articles (15%), and the epoxy-coated steel bar was tested in only one work. Even though several types of concretes and additions are constantly developed, ribbed steel bars seem to be the best option when employed in RC structures built with CDW-concrete, since they present mechanical properties with lower values compared to natural aggregate concrete [15], which negatively influences the bond strength. Furthermore, it is worth mentioning that the influence of the reinforcement material, e.g., carbon or stainless steel, on the bond capacity developed is more relevant for smooth specimens than for ribbed ones, which is explained by the chemical adhesion mechanism that governs the bond behavior of smooth specimens [57].

Regarding the bar diameter, Table 4 shows that diameters between 8 mm and 30 mm were employed in the selected studies. The most frequent value was 16 mm, used in ten manuscripts (50%), followed by 12 mm in seven (35%). Considering that the bar diameter directly influences the adherence with the concrete, because of the contact area between the two materials, the tendency that bars with a larger diameter are used in CDW-concrete, compared to thin bars, is evident. In the selected bibliography, for example, only two articles (10%) tested bars with a diameter of 8 mm, and both reported lower bond strength for this case. Considering that different results were presented in the selected literature, which may be related to the different concrete types and additions used, and even though there are still no specific standards for thin bars [58], thin bars must be better investigated in future research.

As a consequence of the wide range of diameters studied, the anchorage length, which depends, among other factors, on the diameter (d), varied essentially between 3 d and 12 d in the tests most commonly employed in the literature (pull-out tests). This range indicates that studies have been conducted to better investigate the real influence of the anchorage length on CDW-concrete, but further extensive investigations on this parameter are still necessary, since evaluating it alone may not lead to assertive conclusions. This topic can

be used for further research. In addition to these bar characteristics, several authors have reported that concrete strength is the most important determinant of adherence [59,60], and should also be taken into consideration.

*3.3. What Is the Environmental Perspective Regarding Using CDW Recycled Aggregates in Reinforced Concrete Structures?*

The use of CDW as recycled aggregate in concrete still faces some obstacles, because they are very heterogeneous materials and have mechanical properties with lower values compared to natural aggregates [14,15]. In addition, they usually contain residual mortar adhering to their surface, which makes them more porous and, consequently, they have higher water absorption capacity. These points make it difficult to use them, especially in cement-based materials [61,62]. Therefore, they are commonly destined for landfills, which in turn produces greenhouse gases [16,17].

Still, advances are noted in studies on the partial or complete replacement of natural aggregates by recycled ones in concrete. The Brazilian standard NBR 15116 [18] allowed up to 20% replacement in its latest revision, in 2021. Just as a comparison, in terms of contaminants (<1%), some international standards are more restrictive than the Brazilian standard [63]. This fact represents a major advance from an environmental perspective, especially for the Brazilian scenario.

Moreover, considering the large $CO_2$ emissions involved in the production of most conventional concretes, the use of CDW-concrete has proven to be a more sustainable alternative, with some recent studies indicating its capacity to capture $CO_2$ from the atmosphere. Kaliyavaradhan and Ling [8], for instance, studied the potential of $CO_2$ sequestration through CDW and stated that because of CDW's alkalinity, they have a great potential in capturing $CO_2$ by the formation of stable carbonate minerals, just like calcium carbonate ($CaCO_3$). In addition, these authors concluded that smaller particle size samples are more effective in capturing $CO_2$ and that the carbonation process increases the density and mechanical strength and reduces the water absorption and drying shrinkage of recycled aggregates. This makes them suitable for the construction industry and some estimations confirm the capacity to capture 270 kg of $CO_2$ by the carbonation process in the concrete wastes. Zhang et al. [9], in turn, developed a model to estimate concrete debris generation and measure the $CO_2$ mitigation potential by recycling concrete wastes. They concluded that the $CO_2$ sequestration by the CDW has the potential to capture approximately 457.7 Mt of carbon dioxide until 2035, with cumulative mitigation achieving 2968 Mt of $CO_2$ from 2018 until 2035. In addition, the $CO_2$ captured by the debris improved the physical properties of the CDW recycled aggregates and decreased the curing time of the CDW-concrete. These authors also noted that 3.1 billion tons of concrete debris were generated all over the world, which could have captured 62.5 Mt $CO_2$ with a profit of 3.3 billion USD according to recycled market values in the year. With this perspective, both the aforementioned studies concluded that the $CO_2$ capture by CDW is feasible and may contribute to sustainable development by the reduction of $CO_2$ in the atmosphere, increasing the properties of CDW-concrete and making them more attractive to the construction industry. As a consequence, this may lead to the development of the recycling industry and reduce the impacts of CDW in landfills.

Given these points, it is worth mentioning the need to further investigate the use of CDW recycled aggregates in RC structures, so that the limit established by current standards can be safely increased. In this regard, steel–concrete bonding plays a fundamental role. Therefore, studies on the mechanical properties, durability, and adherence of CDW-concrete with reinforcement bars, including alternative bars to steel ones, such as carbon fiber-reinforced polymer (CFRP) bars, are suggested.

## 4. Conclusions

A systematic review of the literature on the bonding of steel bars in concretes produced with recycled aggregates was performed in this paper. The SREE method was efficient and

resulted in 20 relevant articles fully aligned with the research topic, whose analysis led to the following conclusions:

i. A replacement content of natural aggregates by recycled ones that would lead to excellent results of bond strength between steel bars and CDW-concrete was not identified in the selected literature. On the contrary, most of the manuscripts indicated the worsening of bond strength compared to conventional concrete;

ii. Carbon steel bars with a ribbed surface were the most employed in the selected bibliography and they seem to be the best option when employed in RC structures built with CDW-concrete. Furthermore, diameters between 8 mm and 30 mm were employed in the studies, with bars with diameters smaller than 8 mm being employed in only two articles. This wide range of diameters led to varied anchorage lengths used in pull-out tests, but the influence of this parameter on steel–concrete bonding still needs to be better understood. This topic can be used for further research;

iii. From an environmental standpoint, the use of CDW as recycled aggregates in concrete needs to be encouraged, since remarkable advances have been obtained in recent studies. Specifically, it can reduce incorrect disposal of CDW and discourage the use of landfills, which emit greenhouse gases, and also capture $CO_2$ from the atmosphere, an eco-friendly alternative.

These conclusions reveal that studies on the bonding of steel bars in CDW-concrete are still necessary and therefore encouraged by the authors. It is worth mentioning that these findings are limited to the analysis of the selected bibliography. Other limitations are the type and degree of confinement, the bar's yield stress, the water/cement ($w/c$) ratio, and the concrete age, which were not considered in the review and also need future investigations. Based on the above-mentioned findings, further investigations should focus on the real influence of recycled aggregate type and replacement content, bar diameter, anchorage length, and CDW's potential to capture $CO_2$.

**Author Contributions:** Conceptualization, E.D.R.; methodology, E.D.R., H.C.G. and R.C.d.A.; formal analysis, E.D.R. and H.C.G.; investigation, E.D.R.; resources, A.C.d.S.B.; data curation, E.D.R. and H.C.G.; writing—original draft preparation, E.D.R., H.C.G., R.C.d.A., F.S.J.P. and A.C.d.S.B.; writing—review and editing, R.C.d.A., F.S.J.P. and A.C.d.S.B.; visualization, R.C.d.A. and F.S.J.P.; supervision, F.S.J.P. and A.C.d.S.B.; project administration, A.C.d.S.B.; funding acquisition, A.C.d.S.B. All authors have read and agreed to the published version of the manuscript.

**Funding:** This research received no external funding.

**Institutional Review Board Statement:** Not applicable.

**Informed Consent Statement:** Not applicable.

**Data Availability Statement:** All the data in the analyses of this study have been listed in the paper.

**Acknowledgments:** We would like to acknowledge the Centro Federal de Educação Tecnológica de Minas Gerais (CEFET-MG) and Fundação de Amparo à Pesquisa do Estado de Minas Gerais (FAPEMIG) for funding this research.

**Conflicts of Interest:** The authors declare no conflict of interest.

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
