# Peer review of "Bonding of Carbon Steel Bars in Concrete Produced with Recycled Aggregates: A Systematic Review of the Literature"

_carbon, 2022_

Round 1

Reviewer 1 Report

Bonding of Carbon Steel Bars in Concrete Produced with Recycled Aggregates: A Systematic Review of the Literature

Manuscript Number:

In the present paper authors provide an overview of the current progress research in reducing Co2 by using the demolition waste (CDW) as a replacement in cement. However, the paper requires some improvement before it can be recommended for publication, it is proposed to re-submit a thoroughly revised version of the manuscript, considering the following comments.

1.     Title is ok 2.     Overall recommendation should be reported in one sentence at the end of the abstract 3.     Lin 18 in Abstract “SREE” should be defined 4.     The authors should overview the recent progress made in the relevant area in the past two years or so such as: https://doi.org/10.1016/j.conbuildmat.2005.08.008; https://doi.org/10.1016/j.jmrt.2020.07.092 ·       etc. ·       In introduction section, please provide the flow chart of this study. It's will make follow this study easy for readers. 5.     Emphasizing the importance of research in introduction 6.     The paper is well written and it is easy to follow, only the authors needs to go thoroughly revised version to correct the typo-mistake. 7.     Conclusion part not well organized, please revise this part

8.     Author should highlight the assumptions and limitations and future research direction of the study.

Reviewer 2 Report

Please further elaborate on the novelty of this review work in abstract.

The presented introduction is pretty modest. Please include a brief but critical review regarding the conducted research studies in the introduction.  It is recommended to add a section “research significance” and highlight the main contribution of the offered review.

You may remove the figure 6. The nationally of the authors would not play an important role in the offered outcomes.

Please add a comprehensive analysis regarding the bonding effect of carbon reinforcement embedded in different recycled materials.

Similarly, please opt out of journal names as it is irrelevant to the reported results.

Please add a compelling discussion on the main parameters that can significantly affect the result of the conducted research works.

Please further highlight the shortcomings in the reviewed studies and include recommendations for future research.

Round 2

Reviewer 2 Report

N/A